# Message Source Credibility and E-Cigarette Harm Perceptions among Young Adults

**DOI:** 10.3390/ijerph19159123

**Published:** 2022-07-26

**Authors:** Donghee N. Lee, Elise M. Stevens

**Affiliations:** Department of Population and Quantitative Health Sciences, Division of Preventive and Behavioral Medicine, UMass Chan Medical School, Worcester, MA 01655, USA; elise.stevens@umassmed.edu

**Keywords:** electronic cigarettes, vaping, communication, health education messaging

## Abstract

This study examined the effect of message source credibility on e-cigarette harm perceptions among U.S. young adults. An online experimental study was conducted where young adults (*n* = 302, Mage = 23.7) were randomized to an e-cigarette public health education message from an expert or a peer young adult. Then, participants answered questions about their perceptions about the message source and e-cigarettes. Results suggest that young adults rated experts as a more credible source (vs. peer) (b = −0.39, SE = 0.15, 95% CI [−0.67, −0.10], *p* < 0.01). Young adults reported greater perceived credibility of the expert message (vs. peer), which was associated with increased e-cigarette harm perceptions. Increased perceived source credibility mediated the association of increased e-cigarette absolute harm perceptions from viewing an expert message (b = −0.11, SE = 0.04, 95% CI: −0.20, −0.02). Source credibility should be considered when designing e-cigarette education messages for young adults.

## 1. Introduction

Young adults report the highest rates of e-cigarette use, with 9.3% reporting using “every day” or “some days” [1]. Young adults are especially vulnerable to the health harms of e-cigarettes because their brains are still in development [2], and use of e-cigarettes can potentially lead them to use more harmful products in the future, such as combustible cigarettes [3,4,5]. Low risk perceptions of e-cigarettes are one of the reasons young people report using e-cigarettes [6,7,8]. To this end, targeted national campaigns have focused on increasing harm perceptions about e-cigarettes [9,10,11].

One central factor in increasing persuasiveness of tobacco control messaging is source credibility [12]. Specifically, some individuals trust experts within the field [13,14], while others trust similar or relatable people such as peers when encountering health messages [15,16,17]. Given the mistrust and misinformation about health information in recent years [18,19,20], it is especially important to understand what sources are most credible to young adults to reduce and prevent risk behaviors, such as the use of e-cigarettes.

Two main types of sources emerge when examining health messages: expert and peer [21]. Expert sources include governmental health agencies [22] and healthcare professionals [13]. Peer sources include people similar to young adults, and this group is often found on social media [15,23]. Although there has not been extensive research on the role of source impacting health decisions about tobacco, past work has shown that trust in the governmental health agencies was associated with greater compliance with health recommendations [24], and lower trust in these agencies was associated with greater e-cigarette use [25,26]. Others have demonstrated that messages from peers were important when obtaining health information [27,28]. 

As such, understanding a trusted source about e-cigarette information may be necessary to effectively increase harm perceptions among young adults. However, existing tobacco control research has focused exclusively on the effect of messages (e.g., attitudes, intentions, and use behaviors) and has largely overlooked the role of message source [12]. Furthermore, the existing literature on e-cigarette information sources has been drawn from the Health Information National Trends Survey (HINTS), and thus the sample has been largely skewed to older populations [25,26]. The purpose of this study was to test whether young adults perceived an expert or peer to be a more credible source of e-cigarette health information, and which source could better help increase e-cigarette harm perceptions among young adults. We hypothesized that perceived source credibility would be associated with e-cigarette harm perceptions, and that perceived source credibility would mediate the association between the message source and e-cigarette harm perceptions. 

## 2. Materials and Methods

### 2.1. Participants

In November 2021, we recruited young adults (*n* = 302) into an online experimental study using a convenience sample from Prolific (Prolific.co), an international crowdsourcing survey platform for behavioral research studies. Participants were eligible if they were between 18 and 30 years old and lived in the United States.

### 2.2. Study Design

Potential participants interested in the study reviewed a brief description about the study on Prolific before being directed to Qualtrics. After providing consent, participants were asked to complete questions pertaining to their e-cigarette and other tobacco use. Next, participants were randomized to see a fact sheet about the health effects of using e-cigarettes that was written by one of two sources (expert vs. peer). Each condition included a statement about the source (e.g., “This message was written by a leading expert in tobacco research at the U.S. Food and Drug Administration;” “This message was written by a young adult just like you”). Terminology and writing styles were adapted to reflect each source type. The expert condition included the term, “electronic vaping products (EVP),” and used technical words to describe contents of e-cigarettes (e.g., toxicants). Alternatively, the peer condition used the term, “vaping,” and used layman terms (e.g., harmful chemicals). After seeing the message, participants rated their opinions about the source and e-cigarettes. After completing the study, participants were compensated $1.95 per Prolific’s policies. All procedures were approved by UMass Chan Medical School’s Institutional Review Board. The messages are available in Appendix A.

### 2.3. Measures

#### 2.3.1. Manipulation Check

We pretested the source types using a small sample (*n* = 17). After seeing each message, participants were asked to rate the perceived likelihood of messages being written by young adults or experts in the field using two questions asking about the likelihood of the message being written by young adults or experts in the field on a scale of 1 (very unlikely) to 5 (very likely). Participants in the expert condition rated the message to be more likely written by experts (M = 3.76) than young adults (M = 2.41), and participants in the young adult condition rated the message to be more likely written by young adults (M = 3.24) than experts in the field (M = 3.06).

#### 2.3.2. Demographics

Participants reported their age, gender, race, and ethnicity (collapsed to Non-Hispanic White and others due to cell sizes), income (collapsed to <$19,999 and >$20,000 due to cell sizes), and education (collapsed to some high school, some college, college graduates due to cell sizes).

#### 2.3.3. E-Cigarette Use

Participants were asked to report their e-cigarette use. Participants were never users if they never tried e-cigarettes, even a puff; participants were categorized as ever users if they had ever tried using e-cigarettes but not in the past 30 days; and current users were those who had used an e-cigarette in the past 30 days.

#### 2.3.4. Perceived Source Credibility

Perceived source credibility was assessed using four questions [29] asking about whether the message source was fair, accurate, told the whole story, and could be trusted on a scale of 1 (not at all) to 7 (very much). Scores were averaged (Cronbach’s α = 0.94).

#### 2.3.5. Absolute E-Cigarette Harm Perceptions

Absolute e-cigarette harm perceptions were assessed using a single item measured on a scale of 1 (not at all harmful) to 5 (extremely harmful). Scales were taken from past work used to measure e-cigarette harm perceptions [30].

#### 2.3.6. E-Cigarette Use Intentions

E-cigarette use intentions was assessed using two questions about their intentions to use an EVP or vape any time during the next year or if it was offered by one of their best friends [31] on a scale of 1 (definitely not) to 4 (definitely yes). Scores were reversed and averaged (Spearman–Brown r = 0.94). 

### 2.4. Statistical Analysis

All analyses were conducted using SPSS v.28. Descriptive statistics were used to calculate distributions of perceived source credibility, absolute harm perceptions, and e-cigarette use intentions. First, we ran three models using multiple regressions to test the association between message source and each of the three outcome variables: perceived source credibility, absolute harm perceptions, and e-cigarette use intentions. According to the source credibility theory, a message from a credible communicator (source) would be associated with greater persuasive effects [21]. Thus, we estimated unadjusted associations of perceived source credibility with absolute harm perceptions and e-cigarette use intentions. Perceived source credibility was significantly correlated with absolute harm perceptions, but it was not associated with use intentions. We then used Hayes PROCESS Macro v.4.0 to estimate the indirect effects of the message source and absolute harm perceptions mediated by perceived source credibility. Models were controlled for age, gender, race, ethnicity, income, and e-cigarette use. See Appendix B for survey questions used for this analysis.

## 3. Results

### 3.1. Participants

On average, participants were 23.7 years old (SD = 3.53), identified mostly as Non-Hispanic White (62.6%), and almost half were male (49.3%). Participants were mostly ever users of e-cigarettes (37.4% vs. 32.1% never users vs. 30.5% current users) and had income below $19,999 (47.7%). Table 1 provides sample characteristics.

#### Participants and Measured Outcomes

We detected significant correlations between demographics and perceived source credibility. Individuals who identified as females (b = 0.36, SE = 0.15, 95% CI [0.07, 0.64], *p* = 0.02) had higher incomes (b = 0.35, SE = 0.15, 95% CI [0.06, 0.63], *p* = 0.02), and college graduates (b = 0.32, SE = 0.15, 95% CI [0.03, 0.61], *p* = 0.03) reported higher perceived source credibility (Table 2). 

Participants’ demographic characteristics and e-cigarette use intentions were not significantly correlated. However, e-cigarette status was significantly correlated with e-cigarette use intentions, such that current users reported higher e-cigarette use intentions (b = 1.64, SE = 0.99, 95% CI [1.45, 1.84], *p* < 0.001) (Table 3).

Demographic characteristics and e-cigarette harm perceptions were significantly correlated. Participants who were younger (b = −0.04, SE = 0.02, 95% CI [−0.07, −0.01], *p* = 0.02), identified as female (b = 0.57, SE = 0.10, 95% CI [0.36, 0.77], *p* < 0.001), and had higher incomes (b = 0.23, SE = 0.11, 95% CI [0.02, 0.44], *p* = 0.04) reported higher e-cigarette harm perceptions. E-cigarette status was significantly correlated with absolute harm perceptions, such that current users reported lower absolute harm perceptions (b = −0.45, SE = 0.11, 95% CI [−0.68, −0.23], *p* < 0.001) (Table 4).

### 3.2. Measured Outcomes and Model Descriptions

Participants reported the average scores of perceived source credibility (M = 4.94, SD = 1.28), absolute e-cigarette harm perceptions (M = 3.84, SD = 0.92), and e-cigarette use intentions (M = 2.03, SD = 1.09). 

#### 3.2.1. Associations of Message Source and Outcomes

Message source and perceived source credibility was significantly correlated, with the expert message being perceived as more credible than the peer message (b = −0.38, SE = 0.14, 95% CI [−0.67, −0.10], *p* < 0.01). However, message source was not significantly correlated with absolute harm perceptions (b = −0.06, SE = 0.11, 95% CI [−0.27, 0.15], *p* = 0.59). Additionally, message source was not significantly correlated with e-cigarette use intentions (b = 0.06, SE = 0.13, 95% CI [−0.19, 0.31], *p* = 0.62). The interaction between e-cigarette use status and message source on outcomes was not statistically significant. 

#### 3.2.2. Mediated Association of Message Source and Absolute Harm Perceptions

The mediated association consisted of three models. The first model estimated the association between message source (explanatory variable: X) and perceived source credibility (response variable: M), the second model estimated the association between perceived source credibility (explanatory variable: M) and e-cigarette harm perceptions (response variable: Y), and the third model estimated the association between message source (explanatory variable: X) and e-cigarette harm perceptions (response variable: Y). The expert source message was associated with increased perceived source credibility (b = −0.39, SE = 0.15, 95% CI [−0.67, −0.10], *p* < 0.01). Increased perceived source credibility was associated with increased absolute harm perceptions (b = 0.27, SE = 0.04, 95% CI [0.20, 0.35], *p* < 0.001). The direct effect of message source on e-cigarette absolute harm perceptions was not statistically significant (b = 0.09, SE = 0.09, 95% CI [−0.08, 0.28], *p* = 0.32). Finally, we estimated the indirect effect of message source (X) on e-cigarette absolute harm perceptions (Y) mediated by perceived source credibility (M). The indirect effect was statistically significant, such that the expert source message was associated with increased e-cigarette harm perceptions mediated by increased perceived source credibility (b = −0.11, SE = 0.04, 95% CI [−0.20, −0.02]). See Figure 1.

## 4. Discussion

Findings from our study illuminate the importance of perceived source credibility on increasing absolute e-cigarette harm perceptions among young adults. Overall, the e-cigarette education message written by an expert was associated with increased perceived source credibility compared to the message written by a peer young adult. While the expert message was not associated with e-cigarette harm perceptions or use intentions, it was associated with increased e-cigarette harm perceptions when the message source was perceived as credible. These findings raise two important elements critical to strengthening tobacco control campaigns and research: (1) identifying audience characteristics, and (2) using a credible source to increase harm perceptions, which can predict behavioral changes [6,8,32].

First, we found that young adults perceived a federal health expert (i.e., FDA) as a more credible source of e-cigarette information than a young adult peer, and that increased perceived source credibility was associated with increased e-cigarette harm perceptions. This contrasts with prior research that had found trust in governmental health agencies or public health groups was not associated with harm perceptions [25,26]. One plausible explanation is differences in sample characteristics, as prior research largely consisted of an older adult sample, who exhibit different tobacco use characteristics compared to younger adults [1]. Interestingly, we did not find a significant association between perceived source credibility and e-cigarette use intentions. It is likely that a repeated exposure to multiple messages is necessary to change long-established beliefs about risk behaviors [33,34]. Overall, results of our study provide information about young adults’ perceptions toward an e-cigarette education message source, which is necessary to inform the development and delivery of targeted public health campaigns for young adults.

Second, in this study, we found that the direct effects of message source on harm perceptions and e-cigarette use intention were not significant. This indicates that e-cigarette education messages can influence risk beliefs only when young adults perceive the source as credible. Initial evidence from national targeted anti-e-cigarette campaigns has shown that increasing knowledge and risk beliefs [9] is critical for changing e-cigarette use behaviors [10]. However, increased attention toward the development of effective anti-e-cigarette campaigns is necessary as e-cigarette susceptibility and use behaviors among young adults remains disproportionately high [35]. Our findings suggest that perceived source credibility may be an underlying mechanism influencing young adults’ harm perceptions from viewing these campaign messages, with a potential to influence their e-cigarette use behaviors [6]. As such, our results add important knowledge about the role of perceived source credibility and its central influence in e-cigarette risk beliefs to the tobacco control literature [12]. Additionally, the differences in the expert and peer messages from our study underscore the need to move away from the one-size-fits-all approach toward the development of e-cigarette education messages using diverse sources to influence e-cigarette use behaviors among young adults. 

Finally, we found sociodemographic correlates of e-cigarette harm perceptions. Individuals with lower incomes and current e-cigarette users reported lower e-cigarette harm perceptions, which potentially suggests that specific social groups are more vulnerable to using e-cigarettes. Specifically, the current pattern of e-cigarette harm perceptions may reflect higher tobacco use rates among the socioeconomically disadvantaged population. Development of targeted public health education interventions for specific social groups may help increase e-cigarette harm perceptions and protect them against the tobacco use. Fortunately, in our sample, younger age was associated with greater e-cigarette harm perceptions after viewing the education messages. This indicates a potential to leverage public health education campaigns to increase e-cigarette harm perceptions in young population. In our sample, education levels were not significantly correlated with e-cigarette harm perceptions or use intentions. However, future work may examine how education interacts with other sociodemographic and tobacco use characteristics to improve health equity. 

There are several limitations. First, the messages for each source condition were written using different language styles (plain words vs. technical jargons) to maximize the differences between the two source types, thus it is possible that these message-level differences introduced confounds to our results. However, we checked readability scores, and both message types received the same grade level (Grade 10) and were rated accordingly in the pre-test, making it unlikely that participants’ ability to comprehend the message influenced study results. Moreover, we used convenience sampling to recruit participants, thus our findings may not generalize to all young adult populations. However, prior tobacco experimental studies using crowdsourced convenience samples have shown results comparable to probability samples [36,37]. We also used self-response measures, which can have social desirability bias related to responding to tobacco-related questions [38]. The effect size in our findings were small, as seen in other message design studies [39]. Despite the small effect size, our findings could be practically meaningful when applied to the general population [40]. Additionally, we did not assess participants’ harm perceptions of smoking combusted cigarettes after the message exposure. Prevalence of conflicting e-cigarette information has generated confusion over the health effect of e-cigarettes (vs. cigarettes), resulting in misperceptions that e-cigarettes are more harmful than cigarettes in some people, which has been associated with increased cigarette smoking [41]. However, we assessed both absolute e-cigarette harm perceptions and relative harm perceptions of using e-cigarettes compared to smoking combusted cigarettes after the message exposure, and the participants’ perceptions of e-cigarettes (vs. cigarettes) harm was lower than absolute e-cigarette harm perceptions, indicating that our study was unlikely to increase misperceptions that e-cigarettes were more harmful than cigarettes. Finally, there is a possibility that young adults were using non-nicotine vaping products and did not personally relate to the health effects of e-cigarettes described in the messages. This could have influenced their responses to the messages and e-cigarette use status. Future research should examine how young adults respond differently to nicotine vaping products and non-nicotine vaping products to increase the effectiveness of e-cigarette education messages. 

Nonetheless, to our knowledge, this study is among the first experimental studies to address the gap regarding the effect of message source credibility on e-cigarette harm perceptions among young adults. These findings serve as preliminary information to inform the development and dissemination of e-cigarette prevention and control campaigns. Future studies should qualitatively explore a wider network of e-cigarette information sources, such as community or student organization leaders that may be trusted by young adults. Additionally, the effect of source credibility should also be examined in the social media context, given the association of lower e-cigarette harm perceptions and social media prevalence among young audiences [15].

## 5. Conclusions

Our study found that an e-cigarette education message from an expert was associated with increased harm perceptions among young adults, because the source was perceived as credible compared to a peer message. Importantly, in this study, young adults were more likely to perceive an expert as a credible source of e-cigarette education messages than a peer young adult. Findings from our study can inform public health officials to strengthen e-cigarette prevention and control campaigns targeting young adults.

## Figures and Tables

**Figure 1 ijerph-19-09123-f001:**
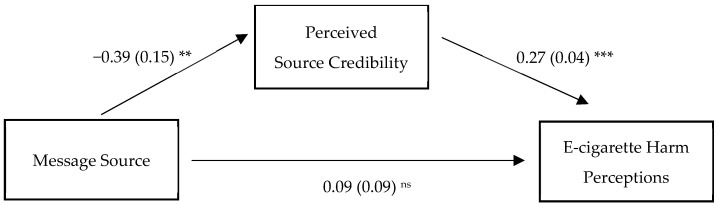
Indirect pathway from message source to e-cigarette harm perceptions mediated by perceived source credibility (*n* = 302) ^1^. ^***^
*p* < 0.001, ^**^
*p* < 0.01, ns: not statistically significant. ^1^
*n* = 302 young adults were randomized to view one of two message sources. Hayes PROCESS Macro was used to model mediated association between source type and outcome variables. The expert source condition was the reference group. The expert source (vs. peer) was associated with increased perceived source credibility. The source conditions were dummy coded into binary variables, with the expert group representing 0 and the peer group representing 1. Increased perceived source credibility was associated with increased e-cigarette harm perceptions. Unstandardized coefficient and standard error denote each path. Covariates included sociodemographic variables and e-cigarette use status.

**Table 1 ijerph-19-09123-t001:** Sample characteristics, *n* = 302 young adults, 2021 ^1^.

	N (%) or Mean (SD)
Race and ethnicity	
Non-Hispanic White	196 (64.9%)
Other ^2^	106 (35.1%)
Age (years)	23.7 (3.53)
Income ^3^	
$0-$19,999	143 (47.7%)
>$20,000-$39,999	67 (22.3%)
$40,000-$49,999	26 (8.7%)
$50,000 and above	64 (21.3%)
Gender ^4^	
Male	148 (49.3%)
Female	140 (46.7%)
Other ^#^	12 (4.0%)
Education ^5^	
Some high school or high school graduate	55 (18.4%)
Some college	120 (40.1%)
College graduate	124 (41.5%)
E-cigarette use ^6^	
Never use	97 (32.1%)
Ever use	113 (37.4%)
Current use	92 (30.5%)

^1^ Participants were recruited from the online crowdsourcing platform, Prolific. ^2^ Others include Black or African American, Asian, Native Americans, Native American or Alaska Native, Native Hawaiian or Other Pacific Islander, Hispanic or Latino, Do not know/Not sure, or multiple races. ^3^ Participants were asked to provide the total annual individual income from all sources, not including the income of other people in the household in a multiple-choice question. ^4^ Participants were asked to indicate their gender in a multiple-choice question with options consisting of male, female, non-binary, prefer not to answer, and none of the above (open-ended). # Others included non-binary, prefer not to answer, and none of the above. However, we did not receive responses for “prefer not to answer,” and “none of the above” options. ^5^ Some high school or high school graduates include elementary to grade 11, grade 12, or GED; some college includes 1 year to 3 years of college; and college graduates include 4 years or more of college. One response (*n* = 1) indicated preferred not to answer. ^6^ Participants were “never” users of e-cigarettes if they reported never using the product, not even a puff; “ever” users of e-cigarettes if they reported using the product at least once, but not in the past 30 days; and “current” users of e-cigarettes if they reported using the product at least once in the past 30 days.

**Table 2 ijerph-19-09123-t002:** Unadjusted associations of perceived source credibility and demographics.

	b	*p*-Value
Age	−0.02	0.29
Race and ethnicity		
Non-Hispanic White (ref)		
Other ^1^	−0.03	0.87
Income ^2^		
$0-$19,999 (ref)		
>$20,000-$39,999	0.35	0.02
Gender		
Male (ref)		
Female	0.36	0.02
Non-Binary	−0.14	0.70
Education		
Some high school or high school graduate (ref)		
Some college	−0.12	0.44
College graduate	0.32	0.03
E-cigarette use		
Never use (ref)		
Ever use	−0.17	0.25
Current use	−0.05	0.75

^1^ Others include Black or African American, Asian, Native Americans, Native American or Alaska Native, Native Hawaiian or Other Pacific Islander, Hispanic or Latino, Do not know/Not sure, or multiple races. *^2^* Income was collapsed to $19,999 and below and $20,000 or above due to cell sizes.

**Table 3 ijerph-19-09123-t003:** Unadjusted associations of e-cigarette use intentions and demographics.

	b	*p*-Value
Age	0.003	0.87
Race and ethnicity		
Non-Hispanic White (ref)		
Other ^1^	−0.22	0.09
Income ^2^		
$0-$19,999 (ref)		
>$20,000-$39,999	−0.12	0.34
Gender		
Male (ref)		
Female	0.32	0.01
Non-Binary	−0.03	0.92
Education		
Some high school or high school graduate (ref)		
Some college	0.08	0.55
College graduate	−0.22	0.08
E-cigarette use		
Never use (ref)		
Ever use	−0.33	0.01
Current use	1.64	<0.001

^1^ Others include Black or African American, Asian, Native Americans, Native American or Alaska Native, Native Hawaiian or Other Pacific Islander, Hispanic or Latino, Do not know/Not sure, or multiple races. ^2^ Income was collapsed to $19,999 and below and $20,000 or above due to cell sizes.

**Table 4 ijerph-19-09123-t004:** Unadjusted associations of e-cigarette harm perceptions and demographics.

	b	*p*-Value
Age	−0.04	0.02
Race and ethnicity		
Non-Hispanic White (ref)		
Other ^1^	−0.05	0.67
Income ^2^		
$0-$19,999 (ref)		
>$20,000-$39,999	0.23	0.04
Gender		
Male (ref)		
Female	0.57	<0.001
Non-Binary	−0.04	0.87
Education		
Some high school or high school graduate (ref)		
Some college	−0.17	0.13
College graduate	0.17	0.11
E-cigarette use		
Never use (ref)		
Ever use	−0.16	0.15
Current use	−0.45	<0.001

^1^ Others include Black or African American, Asian, Native Americans, Native American or Alaska Native, Native Hawaiian or Other Pacific Islander, Hispanic or Latino, Do not know/Not sure, or multiple races. ^2^ Income was collapsed to $19,999 and below and $20,000 or above due to cell sizes.

## Data Availability

Data available upon request.

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
