# Peer review of "Message Source Credibility and E-Cigarette Harm Perceptions among Young Adults"

_ijerph, 2022, doi:10.3390/ijerph19159123_

Round 1

Reviewer 1 Report

The manuscript studies the effect of message source credibility on e-cigarette harm perceptions among US young adults. Based on the online experimental data for people at age 18-30 and living in the US (N=300) in 2021, the authors applied multiple regression models and statistical tests to measure the associations between the independent variables (sociodemographic variables, message source, and e-cigarette use status) and three dependent variables (perceived source credibility, absolute harm perceptions, and e-cigarette use intentions) separately. And then formed a mediated association of message source and absolute harm perceptions through perceived source credibility. As a whole, I found this manuscript well written from creative thinking, and the findings are meaningful for further studies. However, I do have some comments and suggestions that I think need some attention.

1.      Electronic vaping products (EVP)” include nicotine vaping products and non-nicotine vaping products. Not sure if the young adults during the survey knew whether their products contained nicotine or not. If they knew they were using non-nicotine products and the harm of nicotine was emphasized in both expert ad peer fact sheets, they may feel they were safe and gave a low score for e-cigarette harm perceptions or deny they were EVP ever/current use.

2.      In Table 1, the sum of observations by race (Non-Hispanic White vs. Other) is 302 instead of 300. However, the sum of their proportions is 100%. Please make the denominator (N) clear or explain if there is any overlapping between different race categories.

3.      In Table 1, the sum of different genders (male/female/other) is not 100%. Should explain what is “Other” in Gender classification (e.g. does it include transgender/bisexual observations or just missing values). I guess 12 others representing 12% other proportion is wrong, please check the results.

4.      In Table 1, the sum of observations by e-cigarette use status (never/ever/current) is 302 instead of 300. However, the sum of their proportions is 100%. Please make the denominator (N) clear and explain if there is any overlapping between different vaping statuses.

5.      Please provide the survey questions for genders and income, or explain more about these two variables. Especially for income, does it mean annual income or the income during a period by the survey time? Is that a continuous or categorical variable? Why did you choose $20000 as the cut point? How to deal with the missing values when calculating the proportions? The sum of two income levels is 99.4% (not 100%).

6.      In Page 3 line 117, please check if reference 21 (a book written by Hovland at el.) is correct for here. To be more specific, could you summarize the content of the theory or just mention the theory name from the book?

7.      In Page 3 line 132, the $19.999 should be $19,999. Please change the decimal point to a comma.

8.      In Page 4 lines 152-153, “Age, gender, and income were significantly correlated with the absolute harm perceptions, but race was not.” Are these variables significantly correlated with perceived source credibility and e-cigarette use intentions?

It will be better if you can briefly describe and explain the model estimates on these variables. For example, with the increase in ages (or income) from 18 to 30, young adults will give higher/lower scores on absolute harm perceptions. Smoking and vaping disparity is a hot topic, and I am interested in how the harm perceptions and e-cigarette use intentions varied by different ages/genders/races/income levels.

9.      In page 4 line 156, “perceived source credibility (b = -.11, SE = .04, 95% CI: -.20, -.02)” is the only result reported with the confidence interval in the result section, while other results use p values. Please make it consistent with others or explain why here use a 95% CI.

10.  “message source” is an important variable in the models, however, the information on the variable is very limited: “young adults were randomized to view one of two message sources.” Please report the number and proportion of expert and peer sources from the random experimental results. If it is a binary variable, which group represents 0 and which for 1? Otherwise, it’s hard for people to understand the meaning of the beta parameter (b) of the message source.

11.  The description of the models used in section 3.3 is not clear.

To make Figure 1, there should be at least two regression models. The first one measures the association between “message source” and “perceived source credibility” (as the response variable Y). The second one measures the association between “perceived source credibility” (as the explanatory variable X) and “e-cigarette harm perceptions”. When measuring the association between the “message source” and “e-cigarette harm perceptions”, you may use the second model or the third regression model without considering “perceived source credibility”.

Please add more information for different models before introducing the model results, e.g. the number of models you applied, “Perceived Source Credibility” and “E-cigarette Harm Perceptions” were dependent or independent variables or not included in the specific model.

Reviewer 2 Report

This is an interesting manuscript and easy to read.

Hypothesis question and deployed methodology are clear.

Results session needs improvement as presentation of the presentation of acquired data is missing as well as some further analysis of it.

Discussion and conclusions are clear, but could be improved following some further analysis of the data.

Please find below some more specific suggestions for your consideration:

1. Lines 89-91: Demographics

·       Was data on the education level of the participants collected? If so, it would be interesting to also present some results and discussion related to this aspect. If not, it may be noted as a limitation to further explore in a future project.

2. Lines 124-177: Results

·       The presentation of results are missing. You are directly presenting your outcomes. It would be nice to present the absolute numbers or percentage of your results (as you do for demographics) for all the replies you received by the participants to your questions (i.e. for all sub-headlines explained in the methodology), before proceeding with presentation of the outcomes as provided in the sub-session 3.2 and 3.3 of the results. You could also consider to attach the questionnaire you used as annex information.

·       It would be interesting to have some analysis and presentation of the survey results per participant profile. That is to say a more detailed presentation of the replies received from: males vs females; or from high income vs low income participants; or from basic-education vs university-education (if such data exist); or even per age group, e.g. 18-20 year-old vs 20-25 year-old and 26-30 year-old (young adults mentality change rapidly in 18-30 years of age. If data on the precise age of each participant is collected, it would be nice to explore if there are differences in young adults as they grow up). Those results may provide additional information for discussion and perhaps conclusions.

3. Discussion and conclusions

·       Consider amendments in the light of further analysis of the data and the presented results.

·       Line 230: Suggest that your results ‘could be’ (rather than ‘are’) practically meaningful…

·       Lines 231 -240: please review and provide some clarification. First you note that “we did not assess participants’ cigarette harm perceptions after the message exposure” and then you say that “we assessed both comparative to cigarettes and absolute e-cigarette harm perceptions after the message exposure,…”. These two sentences are conflicting.

I hope you will find these suggestions useful.

Kind regards
